# Theoretical Framework for Measuring Cloud Effective Supersaturation Fluctuations with an Advanced Optical System

Ye Kuang[1], Jiangchuan Tao[1], Hanbing Xu[2], Li Liu[3], Pengfei Liu[4], Wanyun Xu[5], Weiqi Xu[6], Yele Sun[6], Chunsheng Zhao[7]

[1] Institute for Environmental and Climate Research, College of Environment and Climate, Jinan University, Guangzhou, Guangdong, China

[2] Experimental Teaching Center, Sun Yat-Sen University, Guangzhou, China

[3] Key Laboratory of Regional Numerical Weather Prediction, Institute of Tropical and Marine Meteorology, China Meteorological Administration, Guangzhou, China.

[4] School of Earth and Atmospheric Sciences, Georgia Institute of Technology, Atlanta, GA, USA

[5] State Key Laboratory of Severe Weather, Key Laboratory for Atmospheric Chemistry, Institute of Atmospheric Composition, Chinese Academy of Meteorological Sciences, Beijing, China

[6] State Key Laboratory of Atmospheric Boundary Layer Physics and Atmospheric Chemistry, Institute of Atmospheric Physics, Chinese Academy of Sciences, Beijing, China.

[7] Department of Atmospheric and Oceanic Sciences, School of Physics, Peking University, Beijing, China.

Correspondence: Ye Kuang (kuangye@jnu.edu.cn)

**Abstract**

Supersaturation is crucial in cloud physics, determining aerosol activation and influencing cloud droplet size distributions, yet its measurement remains challenging and poorly constrained. This study proposes a theoretical framework to simultaneously observe critical activation diameter and hygroscopicity of activated aerosols through direct measurements of scattering and water induced scattering enhancement of interstitial and activated aerosols, enabling effective supersaturation measurements. Advanced optical systems based on this framework allows minute- to second-level effective supersaturation measurements, capturing fluctuations vital to cloud microphysics. Although currently limited to clouds with supersaturations below ~0.2% due to small scattering signals from sub-100 nm aerosols, advancements in optical sensors could extend its applicability. Its suitability for long-term measurements allows for climatological studies of fogs and mountain clouds. When equipped with aerial vehicles, the system could also measure aloft clouds. Therefore, the proposed theory serving a valuable way for both short-term and long-term cloud microphysics and aerosol-cloud interaction studies.

## 1. Introduction

Clouds and fogs play critical roles in weather patterns and climate change, influencing both precipitation and the radiative balance of the Earth's atmosphere. As such, they are central to accurate weather and climate predictions. Despite their importance, representing clouds accurately in atmospheric models remains a significant challenge (Seinfeld and Pandis, 2016) . Supersaturation, defined as the difference between the actual water vapor pressure (e) and the saturation vapor pressure (es) which is typically expressed as a dimensionless quantity (e−es)/es, is a key parameter that links aerosols to clouds through the process of aerosol activation, making it fundamental to cloud physics (Seinfeld and Pandis, 2016). Despite its importance, supersaturation is difficult to measure and remains poorly understood and constrained (Yang et al., 2019). Previous studies have highlighted that other than the mean supersaturation, supersaturation fluctuations also play critical roles in aerosol activation and cloud droplet growth, ultimately influencing the evolution of cloud droplet size distributions (Kaufman and Tanré, 1994;Sardina et al., 2015;Chandrakar et al., 2018;Chandrakar et al., 2020;Shaw et al., 2020). For instance, cloud chamber experiments have shown that supersaturation fluctuations promote aerosol activation and enhance aerosol activity (Shawon et al., 2021;Anderson et al., 2023), particularly when the magnitude of these fluctuations is comparable to the mean supersaturation (Prabhakaran et al., 2020). Both experimental and theoretical analyses suggest that supersaturation fluctuations can broaden cloud droplet size distributions (Chandrakar et al., 2016;Abade et al., 2018;Saito et al., 2019).

Supersaturation fluctuations arise not only from turbulent variations in the temperature and vapor pressure fields but also from the growth and evaporation of droplets, which drive mass and heat exchange between droplets and the surrounding air. As noted by Shaw et al. (2020), measuring supersaturation remains a formidable challenge due to its extreme sensitivity to variations in water vapor pressure and temperature. Although current techniques of water vapor and temperature measurements could not achieve accurately measurements of supersaturation, however, direct measurements of water vapor pressure and temperature were previously used to estimate supersaturation fluctuations, and obtained results have demonstrated that the supersaturation is indeed a fluctuating quantity (Ditas et al., 2012;Siebert and Shaw, 2017).. Currently, cloud and fog supersaturation are typically retrieved from aerosol activation measurements (Ditas et al., 2012) or estimated from vertical velocity measurements and droplet size distribution measurements (Siebert and Shaw, 2017;Cooper, 1989). Supersaturation parameterizations based on vertical velocity are common in models (Abdul-Razzak et al., 1998), while field measurements often rely on aerosol

activation data to investigate supersaturation fluctuations and evolutions in clouds and fogs (Ditas et al., 2012;Hammer et al., 2014;Shen et al., 2018;Mazoyer et al., 2019;Zíková et al., 2020;Wainwright et al., 2021;Kuang et al., 2024). In addition, supersaturations were also estimated using the closure between cloud droplet number and cloud condensation nuclei (CCN) measurements at various supersaturations (Yum et al., 1998;Sanchez et al., 2016;Sanchez et al., 2021;Saliba et al., 2023).

In summary, direct measurements of water vapor pressure and temperature are essential for quantifying supersaturations; however, they are nearly impossible with current technologies. Supersaturation measurements from aerosol and cloud microphysics monitoring often reflect an effective supersaturation that drives aerosol activation, which is indeed critical in cloud physics. The complexity of cloud formation and evolution and the central role of supersaturation in these processes underscore the need for precise measurement and representation of supersaturation. Advancements in measuring and understanding supersaturation are essential for improving the accuracy of models and reducing uncertainties in weather and climate predictions. In this study, we propose a theoretical framework for using optical methods to observe effective supersaturations based on aerosol activation in clouds and preliminarily validated utilizing data obtained from field campaigns. The feasibility of employing an advanced optical system to measure supersaturation fluctuations were also explored and discussed.

## 2. Methods and Materials

### 2.1 Observing effective supersaturations on the basis of κ-Köhler theory

The concept of effective supersaturation was introduced based on aerosol activation measurements (Hudson and Yum, 1997;Hudson et al., 2010), which could be defined as the supersaturation in CCN chamber (CCN activation under constant supersaturation conditions) that resulted in the same aerosol activation fraction with the observed aerosol activation fraction in clouds. Quick fluctuations in supersaturation would result in the effective supersaturation, which directly determined by aerosol activation, differs from the mean supersaturation which is determined by average water vapor content and temperature. However, the concept of κ-Köhler theory is established according to a constant supersaturation scenario, therefore provides a framework for deriving effective supersaturation from aerosol activation measurements in clouds (Petters and Kreidenweis, 2007):

$$S = \frac{D^3 - D_d{}^3}{D^3 - D_d{}^3(1-\kappa)} \cdot \exp\left(\frac{4\sigma_{s/a} \cdot M_{water}}{R \cdot T \cdot D_p \cdot g \cdot \rho_w}\right) \qquad (1)$$

where S is the saturation ratio over an aqueous solution droplet with a diameter of D, $D_d$ is the dry
diameter, $\sigma_{s/a}$ is the surface tension of solution/air interface, $T$ is the temperature, $M_{water}$ is the
molecular weight of water, R is the universal gas constant, $\rho_w$ is the density of water, and $\kappa$ is the
hygroscopicity parameter. The $\kappa$-Köhler theory tells that if the critical diameter of aerosol activation
( $D_a$ ) and corresponding aerosol hygroscopicity parameter $\kappa$ are known, the surrounding
supersaturation can be retrieved based on air temperature measurements and by assuming $\sigma_{s/a}$ the
surface tension of water (as shown in Fig.S1a). Note that $D_a$ and $\kappa$ are not independent with each other,
average $\kappa$ of aerosols with diameter $D_a$ is needed. Previous studies have shown that the reduction in
surface tension (Nozière et al., 2010;Gérard et al., 2016;Ovadnevaite et al., 2017) associated with
surfactants in atmospheric aerosols can affect aerosol activation and, consequently, the derivation of
effective supersaturation. However, if the derivation of $\kappa$ (as done in this study) assumes a constant
water surface tension, the impact of surface tension changes is minimized, as these effects are already
incorporated in the $\kappa$ calculation. Nonetheless, differences in surface tension between supersaturated
and subsaturated conditions (Davies et al., 2019;Petters and Kreidenweis, 2013;Liu et al., 2018), and
their impact on effective supersaturation, still exist. Additionally, prior research has suggested that
slightly soluble components in aerosols can influence $\kappa$ values under both supersaturated and
subsaturated conditions (Ho et al., 2010;Petters and Kreidenweis, 2008;Lee et al., 2022;Han et al.,
2022;Riipinen et al., 2015;Wang et al., 2019;Whitehead et al., 2014). Therefore, $\kappa$ observed under
subsaturated conditions would affect the derivation of effective supersaturation.

139        However, the simultaneous measurements of $D_a$ and $\kappa$ of activated aerosols with diameters

around $D_a$ are indeed challenging. The direct measurements of size-resolved activation ratio (AR) in
clouds are essential for $D_a$ retrievals through the following equation:
$$AR(D_p)=\frac{MAF}{2}\left(1+\mathrm{erf}\left(\frac{D_p\text{-}D_a}{\sqrt{2\pi}\sigma}\right)\right) \qquad (2)$$
Where $D_p$ is the particle diameter, MAF is the maximum activation fraction and $D_a$ is critical
activation diameter, $\sigma$ is associated with the slope of the size-resolved AR curve near $D_a$ and mostly
influenced by the heterogeneous distribution of aerosols near $D_a$ as well as supersaturation fluctuations
(note that not effective supersaturation fluctuations). This formula  was previously proposed by Rose
et al. (2008) to fit the AR measurements and widely used in AR parameterizations (Tao et al., 2018b).
Therefore, it typically requires a unique inlet system and a suite of instruments that measure the aerosol
size distribution of both interstitial and total aerosol populations (Hammer et al., 2014;Zíková et al.,
2020). Consequently, this is rarely done, even in ground fog measurements. Instead, $D_a$ was usually
estimated from aerosol measurements and fog droplet size distributions measurements which
indirectly provides the number concentrations of activated aerosols therefore could be used in
retrieving $D_a$ through assuming that all aerosols larger than $D_a$ are activated (Mazoyer et al.,
2019;Wainwright et al., 2021;Shen et al., 2018) which brings uncertainty in $D_a$ derivations due to that
not all aerosols larger than $D_a$ are activated, because the MAF in Eq.2 does not equal to unit although
usually very close to (Tao et al., 2018b). For the effective supersaturation measured in aloft clouds,
the aerosol number size distributions inside and outside the cloud as well as cloud droplet number
concentrations were used by Ditas et al. (2012) to derive $D_a$, and other approaches were also used
(Gong et al., 2023). The κ values were usually retrieved from size-resolved cloud condensation nuclei
measurements under certain supersaturations (Hammer et al., 2014;Mazoyer et al., 2019) or from
growth factor measurements (Wainwright et al., 2021) or sometime assumed due to the lack of
measurements. The κ of activated aerosols were not directly measured in these studies due to the
difficulty of the direct sampling of activated aerosols as well as subsequent hygroscopicity
measurements.
Two types of supersaturation fluctuations have been previously identified. The first type involves
fluctuations in supersaturation directly governed by water vapor pressure and temperature, as
described by Siebert and Shaw (2017). These fluctuations are linked to turbulence and water phase
changes that influence water vapor pressure and temperature. The second type concerns fluctuations
in effective supersaturation, which are associated with the activation and deactivation processes of
aerosols, as noted by Ditas et al. (2012). The first type of fluctuations dictates the instantaneous growth
and evaporation of droplets, thereby controlling the activation and deactivation of cloud droplets. As
such, the second type of fluctuation is inherently driven by the first type. The theoretical framework
proposed in this study enables the measurement of fluctuations in effective supersaturation.

**2.2 Field measurements**
Kuang et al. [2024] developed an advanced aerosol-cloud sampling system designed to measure
fog and cloud activation processes. This compact, integrated system can automatically switch between
different inlets, including $PM_1$ (particles and droplets with an aerodynamic diameter < 1 μm), $PM_{2.5}$
(particles and droplets with an aerodynamic diameter < 2.5 μm) impactor, and Total Suspended
Particles (TSP, encompassing all particles and droplets) (as shown in Fig. S2). When combined with

instruments that measure aerosol physical, optical, and chemical properties, this system is well-suited for investigating cloud microphysics and chemistry. It was utilized in the AQ-SOFAR campaign, dedicated to studying AQueous Secondary aerOsol formation in Fogs and Aerosols and their Radiative effects in the North China Plain (Kuang et al., 2024).

During this campaign, several radiation fog events were observed, enabling the measurement of size-resolved AR curves, aerosol hygroscopicity as well as chemical compositions of interstitial and activated aerosols within fogs. These measurements provided insights into the evolution of supersaturations (Kuang et al., 2024). Notably, aerosol hygroscopicity was determined using a humidified nephelometer system, located downstream of the inlet system. This system measured multiwavelength scattering coefficients (450 nm, 525 nm, 635 nm) under both nearly dry (RH<20%) and humid conditions (RH~84%), offering aerosol hygroscopicity data based on the optical theory proposed by Kuang et al. (2017). The size-resolved AR curves and aerosol chemical compositions were obtained through the aerosol size distribution and the aerosol mass spectrometry measurements downstream of the inlet system. A schematic of the inlet system and associated instruments is provided in Fig. S1. Further details about the entire experimental setup, size-resolved AR calculations as well as data analysis about mass spectrometer measurements can be found in Kuang et al. (2024).

In addition, the particle number size distributions (PNSDs) in dry state, which range from about 10 nm to 10 μm, were jointly measured by a Twin Differential Mobility Particle Sizer (TDMPS, Leibniz-Institute for Tropospheric Research, Germany) or a scanning mobility particle size spectrometer (SMPS) and an Aerodynamic Particle Sizer (APS, TSI Inc., Model 3321) in six field campaigns conducted on the North China Plain which are detailed in Kuang et al. (2018). The mass concentrations of black carbon (BC) were measured using a Multi-Angle Absorption Photometer (MAAP Model 5012, Thermo, Inc., Waltham, MA USA) or an aethalometer (AE33) (Drinovec et al., 2015) in these field campaigns. Details about these measurements and quality assurance was introduced in Kuang et al. (2018).

**2.3 Method of simulating scattering coefficients of interstital aerosols and activated aerosols**

For each paired PNSD and BC mass concentration, the size distribution of dry-state $PM_1$ was obtained using the following formula (the penetration curve shape from Gussman et al. (2002) was also included for considering the non-ideality cutoff of the impactor, and assuming aerosol density of 1.6 g/cm$^3$ for converting aerodynamic diameter to mobility diameter) :

$\text{PNSD }(D_p)_{PM_1} = \text{PNSD }(D_p) \times R(D_p)$ (3)
Where R(Dp) is the penetration ratio of aerosols as a function of particle diameter $D_p$ of the PM₁
impactor. Further, $\text{PNSD }(D_p)_{PM_1}$ and the BC mass concentration was used to simulate the size-
resolved aerosol scattering coefficients ($d\sigma_{sp}/dlogDp$) at 450 nm, 525 nm and 636 nm that is
consistent with the angular truncation and light source nonideality of Aurora 3000 nephelometer
(Müller et al., 2011), where $\sigma_{sp}$ represents aerosol scattering coefficient. In this Mie calculation, the
shape of black carbon mass size distributions are consistent with the one used in simulations of Kuang
et al. (2017) assuming fractions of BC mass that are externally mixed is 0.5. Details about the Mie
theory calculations can also be found in  Ma et al. (2011) and Kuang et al. (2017).
With given size-resolved AR curve that produced using Eq.2, the size-resolved aerosol scattering
coefficients of interstitial aerosols can be calculated using the following formula:
$d\sigma_{sp,inter}/dlog D_p (D_p) = d\sigma_{sp}/dlog D_p (D_p) \times (1 - AR(D_p))$ (4)
The size-resolved aerosol scattering coefficients of activated aerosols can be calculated using:
$d\sigma_{sp,act}/dlog D_p (D_p) = d\sigma_{sp}/dlog D_p (D_p) - d\sigma_{sp,inter}/dlog D_p (D_p)$ (5)
Scattering coefficients of total aerosol populations (interstitial plus activated) and interstitial aerosols
can be derived through integration of $d\sigma_{sp}/dlog D_p (D_p)$ and $d\sigma_{sp,inter}/dlog D_p (D_p)$.

## 228 3. Theoretical Framework and Concept Design of the Advanced Optical System

### 229 3.1 Theory of Observing Critical Activation Diameter Using Scattering Measurements

The typical shape of size-resolved AR curves observed in atmospheric fogs and clouds is
illustrated in Fig. 1a (Ditas et al., 2012;Hammer et al., 2014;Zíková et al., 2020;Wainwright et al.,
2021;Kuang et al., 2024). In clouds, aerosols can be classified as either activated aerosols, which form
cloud droplets, or inactivated aerosols, which remain as interstitial aerosols. The critical diameter that
distinguishes interstitial aerosols from cloud or fog droplets varies depending on the supersaturation
(Kuang et al., 2024). A diameter of 2.5 μm is typically suitable for surface fogs with relatively lower
supersaturations (<0.1%), while 1 μm is more appropriate for aloft clouds with higher supersaturations
(>0.1%) (Mazoyer et al., 2019;Kuang et al., 2024;Lu et al., 2020). The typical AR curve shows that
most aerosols larger than $D_a$ are activated, while most smaller aerosols remain inactivated. As a result,
the scattering properties, such as size-resolved scattering coefficients (Fig.1a), the scattering Ångström
exponent (SAE) and its wavelength dependence, which are directly related to aerosol size distribution,
differ significantly between interstitial and activated aerosols.

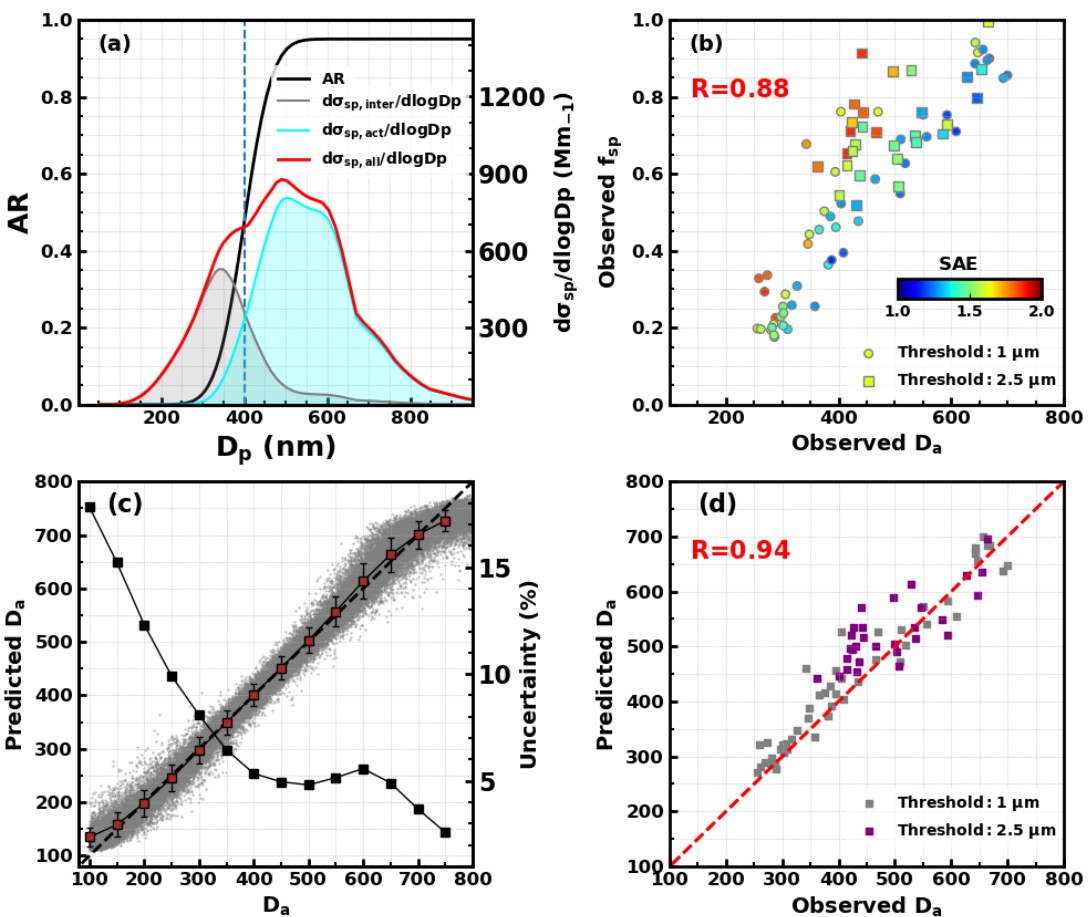

**Figure 1**. **(a)**The typical shape of size-resolved aerosol activation ratio (AR) curve produced using the function of Eq.2, with the Da of 400 nm, the MAF of 0.95 and the $\sigma$ of 30 (as an example). The average PNSD observed in the North China Plain from six campaigns as introduced in Sect2.2 and the example AR curve was used to simulate an example of the size-resolved aerosol scattering ($\sigma_{sp}$) distributions of interstitial and activated aerosols at 525 nm; **(b)** Relations between observed $D_a$ and $f_{sp}$ during the AQ-SOFAR campaign using 1 and 2.5 μm as the threshold of interstital aerosols, with the scatter points are colored with corresponding SAE of total dry state PM$_1$ aerosols. **(c)** Comparisons of all prescribed $D_a$ and predicted $D_a$ values represented by scatter points, they are further binned with interval of 50 nm, averages and standard deviations represented by purple squares and their error bars, black squares represent relative uncertainty of the right axis at each bin; **(d)** The comparisons of $D_a$ retrieved using activation ratio observations and those predicted using scattering observations as inputs of the trained model, dashed lines represent 1:1 lines.

If we focus on PM$_1$ of the total dry aerosol population (the reasoning for this is discussed in Sect.
S1 of the supplement), the scattering fraction of interstitial aerosols in the total dry PM$_1$ population,
defined as $f_{sp} = \sigma_{sp,PM_1,inter}(dry, 525\ nm)/\sigma_{sp,PM_1,all}(dry, 525\ nm)$ , where
$\sigma_{sp,PM_1,inter}(dry, 525\ nm)$ is the scattering coefficient of PM$_1$ interstitial aerosols in a dry state at a
wavelength of 525 nm, and $\sigma_{sp,PM_1,all}(dry, 525\ nm)$ is that of all PM$_1$ aerosols, is likely to be highly
correlated with $D_a$. Generally, the larger the $D_a$, the higher the $f_{sp}$. This relationship was directly
confirmed using $D_a$ and the scattering properties of dry PM$_1$ interstitial and total aerosols during the
AQ-SOFAR campaign, as shown in Fig. 1b, that observed $D_a$ correlates highly with observed $f_{sp}$
(R=0.88). However, at a given $D_a$, $f_{sp}$ can vary significantly, and these variations are closely related
to the SAE of all dry PM$_1$ aerosols, which are mainly determined by aerosol size distribution. In fact,
aside from the size distribution of the total aerosol population that determines SAE, the shape of the
AR curve also plays a significant role in the variations of $f_{sp}$.
The nephelometer measures the aerosol scattering coefficient at three wavelengths, enabling
direct measurements of the SAE for both the total dry-state PM$_1$ aerosols and the interstitial aerosols.
Therefore, the relationship between $f_{sp}$ and $D_a$ can be further constrained by the SAE of interstitial
and activated aerosols, as well as their wavelength dependence. This implies that a simple formulaic
relationship between $f_{sp}$ and $D_a$ may not exist. However, the six scattering parameters
$\sigma_{sp,PM_1,inter}(dry,\lambda)$—at 450 nm, 525 nm, 635 nm, and $\sigma_{sp,PM_1,all}(dry,\lambda)$ at 450 nm, 525 nm, 635
nm—contain both the $f_{sp}$ information and the SAE characteristics of both aerosol groups, thus
potentially be used to accurately retrieve $D_a$. Machine learning techniques, which are well-suited for
handling complex relationships, can be applied to this problem.
This assumption was tested using Mie theory, based on aerosol size distributions sampled during
six campaigns conducted in the North China Plain region (Kuang et al., 2018). For each aerosol size
distribution, we randomly assumed different activation curves using Eq.2. That is, for each PNSD from
those campaigns, the scattering coefficients of submicron interstitial and activated+interstital aerosols
at wavelengths of 450 nm, 525 nm and 635 nm corresponding to nephelometer case under 100 size-
resolved AR scenarios were simulated using the procedure. And each size-resolved AR curve was
produced by using randomly produced $D_a$, σ and MAF as inputs of Eq.2. In the random step, the
range of $D_a$ is 100-700 nm, the range of σ is 1-30, the range of MAF is 0.5-1. In each pair, simulated
$\sigma_{sp,PM_1,inter}(dry,\lambda)$—at 450 nm, 525 nm, 635 nm, and $\sigma_{sp,PM_1,all}(dry,\lambda)$ at 450 nm, 525 nm, 635
nm was the x values of the random forest model, corresponding $D_a$ is the y value of the random forest
model, and the random forest package from Python Scikit–Learn machine learning library
(http://scikit-learn.org/stable/index.html) is used for this purpose. With these configurations, more
than million pairs are simulated. To preliminarily validate this approach, we randomly selected 75%
of the simulated data pairs for training the model, while the remaining 25% were used for validation.
The results, shown in Fig. 1c, indicate that this approach could retrieve $D_a$ with an uncertainty of
less than 10% for $D_a$ larger than 250 nm, and even as low as ~6% for $D_a$ larger than 350 nm. However,
the uncertainty increases as $D_a$ decreases, particularly for $D_a$ smaller than 250 nm. The larger
uncertainty at smaller $D_a$ is since aerosols smaller than 250 nm typically contribute less than 10% to
total scattering in the dry state, making $f_{sp}$ less sensitive to variations in $D_a$. This issue becomes more
pronounced when $D_a$ is less than 100 nm, as aerosols smaller than 150 nm generally contribute
negligibly to total aerosol scattering [Kuang et al., 2018]. This method was further validated using
observations from the AQ-SOFAR campaign. In this validation, $D_a$ values were first predicted using
aerosol scattering observations with the trained model and then compared with $D_a$ values retrieved
from size-resolved AR measurements, as shown in Fig. 1d. It should be noted that the impactor
operates in a sequence of PM$_1$, PM$_{2.5}$, TSP, and then back to PM$_1$, with the flow alternating between
a thermodenuder and bypass every 10 minutes for each inlet. To calculate size-resolved AR curves,
we assumed that aerosol populations remained unchanged during the 30-minute period (based on
comparisons between PM$_1$/PM$_{2.5}$ and TSP inlets), which can sometimes introduce significant
uncertainties in the size-resolved AR calculations. When using PM$_{2.5}$ as the threshold, the much lower
number concentrations of aerosols larger than 400 nm can introduce more uncertainty in $D_a$ retrievals,
partially explaining the lower performance in Fig. 1d when using the PM$_{2.5}$ threshold.
**3.2 Method of observing Hygroscopicity of Activated Aerosols**
Measuring the hygroscopicity $\kappa$ of activated aerosols at the critical activation diameter $D_a$ under
varying supersaturations is challenging, not only due to technical limitations but also because of the
inherent variability in $D_a$. Kuang et al. (2017) introduced a novel optical method for observing aerosol
hygroscopicity by using the aerosol light scattering enhancement factor $f(RH)$ that associated with
aerosol hygroscopic growth. This method is particularly suitable for the objectives outlined here. The
method requires SAE and light scattering enhancement factors $f(RH)$ of activated aerosols as inputs,
and retrieved $\kappa$ can be termed as $\kappa_{act,f(RH)}$ which represents the overall hygroscopicity of activated
aerosols and can be understood as the average $\kappa$ of activated aerosols with the scattering contribution
of each aerosol particle as the weight (Kuang et al., 2020). The scattering coefficients of activated
aerosols at multiwavelength can be calculated as $\sigma_{sp,PM_1,act}(dry,\lambda) = \sigma_{sp,PM_1,all}(dry,\lambda) -$
$\sigma_{sp,PM_1,inter}(dry,\lambda)$, therefore corresponding SAE can be obtained. The $f(RH)$ of activated aerosols
at 525 nm can be calculated as the following:
$$f(RH)_{act} = \frac{\sigma_{sp,PM_1,all}\,(RH,525\,nm) - \sigma_{sp,PM_1,inter}(RH,525\,nm)}{\sigma_{sp,PM_1,all}(dry,525\,nm) - \sigma_{sp,PM_1,inter}(dry,525\,nm)} \qquad (6)$$
During the AQ-SOFAR campaign, a humidified nephelometer system consisting of two
nephelometers—one measuring aerosol scattering in the dry state and the other at a fixed RH of 84%—
was placed downstream of the PM₁ impactor. This setup allows for the humidification of dry-state
interstitial aerosols and total aerosol populations to a high RH (e.g., above 80%), facilitating the
required measurements, therefore severs one choice. The Retrieved $\kappa_{act,f(RH)}$ under different $D_a$
conditions are shown in Fig.2a, demonstrating significant variations in $\kappa_{act,f(RH)}$ and its variations
need to constrained. Also, the derived $\kappa_{act,f(RH)}$ are compared those estimated from aerosol chemical
compositions measurements ($\kappa_{act,chem}$, details about calculation methods can refer to Kuang et al.
(2020)), as shown in Fig.2b and in general agree. Note that the mass spectrometer could not identify

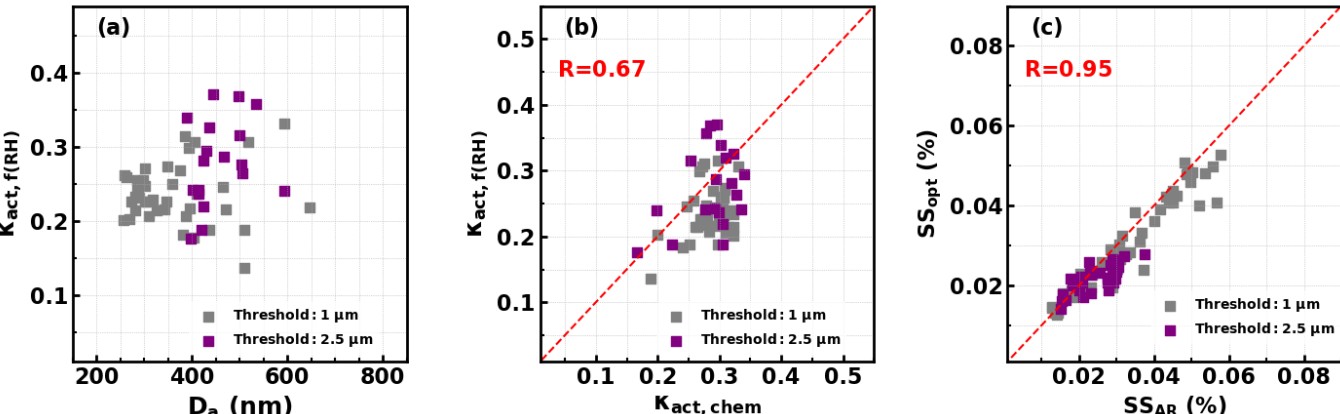

**Figure 2**. **(a)** Retrieved $\kappa_{act,f(RH)}$ under different $D_a$ conditions; **(b)** Comparison between $\kappa$ of activated aerosols retrieved from the optical method ($\kappa_{act,f(RH)}$) and estimated from aerosol chemical composition measurements ($\kappa_{act,chem}$) ;**(c)** Comparisons between effective supersaturations (SSs) derived from size-resolved AR measurements as well as $\kappa_{act,chem}$ ($SS_{AR}$) and from the optical measurements ($SS_{opt}$). Dashed red lines represent 1:1.

all aerosol components, and assumptions about the mixing rule as well as densities of components
would bring uncertainties (Kuang et al., 2021). The comparisons between effective supersaturations
derived from size-resolved AR measurements as well as $\kappa_{act,chem}$ and from the optical method are
shown in Fig.2c. On average, 0.002% of SS bias are observed due to the bias of $D_a$ which associated
more with assumptions made in $D_a$ retrievals as previously discussed. As demonstrated by Kuang et
al. (2024), for the fog case in the campaign, the threshold of 2.5 μm should be used, however, does not
affect the comparisons here.
Qiao et al. (2024) developed an advanced outdoor nephelometer system that measures aerosol
dry scattering coefficients and scattering coefficients at nearly ambient RH without the need for
humidifying the sample air by placing the entire nephelometer system in ambient air, with the
instruments protected by a specially designed enclosure. This innovative design offers new insights
into the hygroscopicity measurements of activated aerosols. Under cloud conditions, where the
ambient RH is close to 100%, aerosol scattering under subsaturated conditions can be measured
directly by applying heater.

## 3.3 Concept Design of the Advanced Optical System for Measuring Effective Supersaturations

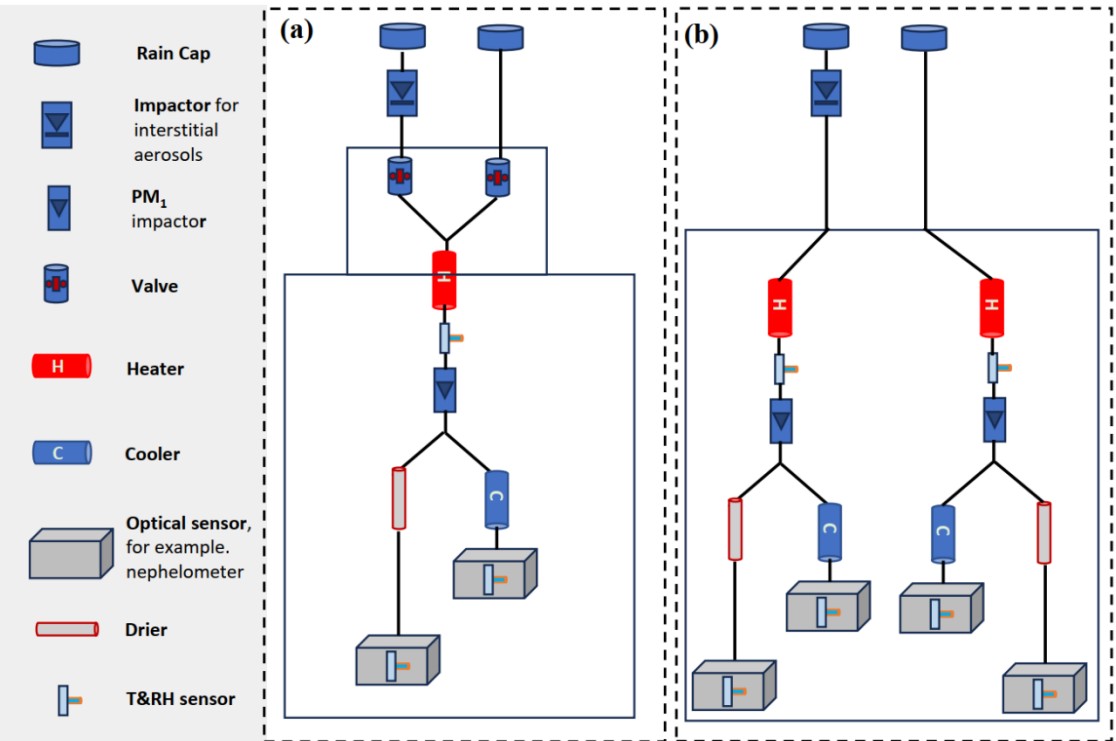

**Figure 3**. Concept design of the advanced optical system with different number of optical sensors, **(a)** using two nephelometers or other optical sensors; **(b)** using four nephelometers or other optical sensors. The heater upstream of the sample is used to reduce the relative humidity (RH) to below 60%, ensuring the evaporation of most of the water content, to make sure the consistency of needed PM$_1$ cut. The cooler upstream of the 'wet' nephelometer increases the sample RH to approximately 90%, allowing hygroscopicity measurements under conditions close to supersaturation.

Based on the proposed optical methods for measuring $D_a$ and $\kappa_{act,f(RH)}$, a conceptual design for
outdoor instruments capable of measuring effective supersaturation with relatively high time
resolution can be envisioned, as shown in Fig. 3a. The aerosol-cloud sampling system includes two
inlets: one equipped with a PM$_1$ or PM$_{2.5}$ impactor (depending on cloud type) to sample interstitial
aerosols, and another with a TSP inlet to sample both interstitial aerosols and cloud/fog droplets. A
PM$_1$ impactor is placed downstream of the inlet system, where the RH of the sample air is reduced to
70% (as discussed in Sect. S1 of the manuscript) through heater. Downstream of the PM1 impactor,
the sample flow is split into two streams: one is further dried to an RH below 10% before aerosol
scattering coefficients are measured by the "dry" nephelometer, and the other is passed through an
intelligent cooler to ensure the sample RH in the "wet" nephelometer remains close to 90%. The
sample air is automatically switched between the interstitial inlet and the TSP inlet at set intervals,
such as one minute for each inlet, enabling minute-level measurements of effective supersaturations.
While the nephelometer can output scattering measurements every second, reliable data can only be
achieved at intervals of around 30 seconds (exact values can be determined through future testing) due
to the residence time of aerosols in the nephelometer and potential light source instability. If four
nephelometers are available, a more advanced optical system can be designed (Fig. 3b) that does not
require switching between the interstitial inlet and the TSP inlet. Instead, two nephelometers would be
placed downstream of the interstitial inlet and two downstream of the TSP inlet, enabling higher time
resolution effective supersaturation measurements. Other types of optical instruments exist that can
achieve stable second-level aerosol scattering or extinction measurements with a stable laser light
source [Moise et al., 2015; Zhou et al., 2020]. Therefore, with the development of suitable optical
instruments, it may be possible to achieve second-level effective supersaturation measurements.
**4. Discussions on Limitations and Advantages**

356        The proposed theoretical framework enables simultaneous measurements of $D_a$ and $\kappa$ for

activated aerosols, leveraging the high time resolution of optical instruments to potentially provide
second-level measurements of supersaturation. However, several limitations should be discussed and
might be improved upon: **(1) Shape of size-resolved AR curve:** Cloud chamber studies have shown
that supersaturation fluctuations can lead to the coexistence of particles with the same critical
supersaturation as both interstitial aerosols and cloud droplets (Shawon et al., 2021). This results in
size-resolved AR curves deviate more from stepwise shape, a phenomenon also observed in some field
measurements (Henning et al., 2004;Mertes et al., 2007). Despite this, a critical diameter $D_a$ still exists,
and such non-ideal curves can be treated a high standard deviation $\sigma$ in the activation error function
(Eq. 2), which does not fundamentally undermine the proposed framework, however, should be further
checked for different cloud types. **(2) Measurement of $\kappa$:** Although the framework measures the
overall $\kappa$ of activated aerosols, the $\kappa$ needed for supersaturation calculations is that of aerosols near
$D_a$ ($\kappa_{D_a}$). For $D_a > \sim 200$ nm, the derived $\kappa_{act,f(RH)}$ can provide a first-order estimate of $\kappa_{D_a}$, based on
observed size-dependent characteristics of $\kappa$ values (Liu et al., 2014;Shen et al., 2021;Wang et al.,
2024), though more comprehensive evaluations are needed. Additionally, $\kappa$ measured under
subsaturated conditions differs from that under supersaturated conditions (Tao et al., 2023) might also
bring some uncertainties. However, as shown in Fig. S1b, even a bias of 0.1 in $\kappa$ only result in a ~0.01%
bias when SS is ~0.1% and a ~0.005% bias when SS is ~0.05% in supersaturation retrievals, making
the first-order estimates of $\kappa_{D_a}$ from optical measurements generally suitable for supersaturation
observations. **(3) Limitations in $D_a$ Retrievals:** Current techniques using aerosol scattering
measurements at visible wavelengths (e.g., nephelometers) are reliable only for $D_a$>100 nm as shown
in Fig.1a, limiting effective supersaturation measurements to less than 0.21% (assuming a typical $\kappa$ of
0.3). This restriction makes the technique most applicable to fog and stratus or stratocumulus cloud
measurements. However, incorporating scattering measurements at ultraviolet wavelengths could
improve sensitivity to smaller $D_a$ and lower $\kappa$, enabling measurements in conditions with higher
effective supersaturation and a broader range of cloud types in the future.

382       The uncertainty in effective supersaturation observations using this framework primarily arises

from the uncertainties in deriving $D_a$ and $\kappa_{D_a}$. The uncertainty in $D_a$ observations using the Aurora
3000 nephelometer as the optical sensor under varying conditions is detailed in Fig. 1c. Factors
affecting the accuracy of $\kappa_{D_a}$ include: (1) the size dependence of $\kappa$ of activated aerosols; (2)
uncertainties related to surface tension, slightly soluble components, and other factors that lead to
differences in $\kappa$ differences under subsaturated and supersaturated conditions. Based on previous
studies on the size dependence of $\kappa$ (Peng et al., 2020) and the differences between subsaturated and
supersaturated conditions (Whitehead et al., 2014;Liu et al., 2018;Tao et al., 2023), a 50% uncertainty
(three times the standard deviation) was assumed in the derivation of $\kappa_{D_a}$ for the uncertainty analysis.
Using this approach, the uncertainty in effective supersaturation measurements, estimated through the
Monte Carlo method, is shown in Fig. 4. The analysis indicates that applying this framework with the
Aurora 3000 nephelometer as the optical sensor results in an uncertainty of approximately 5%. The
precision of effective supersaturation measurements is directly linked to the accuracy of the optical
sensor's scattering signal. For example, the Aurora 3000 has an accuracy of 1 Mm$^{-1}$, which leads to
different levels of precision in $D_a$ and hygroscopicity measurements depending on the scattering signal
strength. If the scattering signal from the total aerosol population is 100 Mm$^{-1}$, the precision of the
observed interstitial aerosol scattering fraction $f_{sp}$ is about 1%. Based on the relationship between $f_{sp}$
and $D_a$ shown in Fig. 1b, this leads to a precision of approximately 3 nm for $D_a$, which results in an
effective supersaturation precision of ~0.01% when supersaturation is near 0.2%, or ~0.0002% when
supersaturation is near 0.02%. However, if the scattering signal is lower (e.g., 10 Mm$^{-1}$), a bias of 1
Mm$^{-1}$ could result in effective supersaturation bias to as much as ~0.07% when supersaturation is near
0.2%, making the measurements unreliable. In summary, while the proposed framework demonstrates
the feasibility of observing effective supersaturation with an advanced optical system, the accuracy

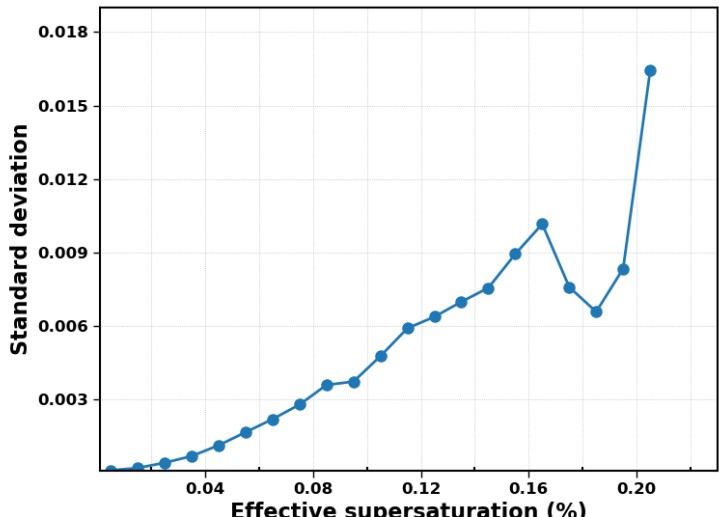

**Figure 4**. The standard deviations of effective supersaturations under different effective supersaturation (SS) levels.

and precision depend on the resolution of the optical sensors, the scattering parameters being measured,
and the scattering signal levels of aerosols in clouds. Enhancing the sensor precision to 0.1 Mm$^{-1}$ or
even 0.01 Mm$^{-1}$, and incorporating ultraviolet wavelengths and multiple scattering angles, might
enable high-accuracy supersaturation measurements across a broad range of supersaturation conditions,
especially in cleaner environments.
As mentioned in Sect. 2.1, the theoretical framework proposed in this study is designed to observe
effective supersaturation fluctuations, rather than supersaturation fluctuations themselves. While there
are non-negligible uncertainties associated with observing effective supersaturation using the proposed
theory, the size and hygroscopicity distributions of total interstitial and activated aerosol populations
remain nearly constant when measured with second-scale or shorter time resolution. The parameter
that changes over time is the dynamic exchange between interstitial and activated aerosols.
Consequently, fluctuations in the scattering signals of interstitial and activated aerosols can reflect this
exchange at high temporal resolution. Since effective supersaturation fluctuations result from
underlying supersaturation variations, they could, in principle, provide insights into the causes of these
fluctuations, such as turbulence, though this would require further investigation and endeavor. In
addition, for size-resolved AR, both $\sigma$ and MAF are crucial parameters. However, using scattering
coefficients at just three wavelengths of Aurora 3000 nephelometer is insufficient for accurately
retrieving $\sigma$ and MAF. If $\sigma$ and MAF could be measured more precisely through the extended optical
framework, it would provide deeper insights into supersaturation fluctuations.
Despite these limitations, the proposed theoretical framework represents the first system capable
of directly providing high time resolution measurements of effective supersaturations using a single
instrument. This system is particularly well-suited for surface fog and mountain cloud observations,
and when coupled with aerial vehicles, it could also be employed for measurements in aloft clouds.
The system offers several advantages for cloud and fog measurements: **(1) High-Resolution**
**Supersaturation Measurements:** The system can provide measurements of effective supersaturations
at even a second-level resolution, making it feasible for observing effective supersaturation
fluctuations and supporting investigations into fog and cloud evolution mechanisms. **(2) Long-Term**
**Measurement Capability:** The optical measurements, such as those from the nephelometer system,
are well-suited for long-term observations, making it possible to acquire climatological data on the
variability of fogs and mountain clouds. **(3) Comprehensive Aerosol and Cloud Data:** In addition to
measuring effective supersaturations, the system directly captures the scattering and hygroscopic
properties of both interstitial and activated aerosols. With further algorithm development, it could also
retrieve the number concentrations of available cloud condensation nuclei (CCN) at certain
supersaturations, as well as cloud droplet number concentrations, based on previous studies that have
observed CCN using optical methods (Tao et al., 2018a). **(4) Monitoring Aerosol Hygroscopic**
**Behavior:** The system continuously monitors aerosol hygroscopic behavior under subsaturated
conditions along with the corresponding optical properties. This allows for clear documentation of the
formation and dissipation of fog/cloud events, as well as the variation in aerosol optical and
hygroscopic properties. Overall, the datasets generated by this system are well-suited for in-depth
investigations of cloud physics and aerosol-cloud interactions. This system has the potential to
significantly advance fundamental research on clouds and fogs. However, further theoretical studies
are needed to refine and optimize this type of system.






**Financial Supports**. This work is supported by National Natural Science Foundation of China (42175083, 42175127), and the Fundamental Research Funds for the Central Universities.

**Competing interests**. The authors declare that they have no conflict of interest.

**Data Availability**. All data presented in Figures of this manuscript are freely available at Kuang, Y. (2024), and more specific data will be made available on request.

**Author contribution**. YK conceived the theoretical framework and wrote the manuscript. JT, HX, L L, WX and WeX participated the field campaign and conducted measurements of aerosol chemical and physical properties. YS, PL, CZ reviewed and commented on the paper.

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
