# Peer review of "Theoretical Framework for Measuring Cloud Effective"

_EGUsphere, 2024_

## Referee Comment (RC1)

**Comments on "Theoretical Framework for Measuring Cloud Effective Supersaturation Fluctuations with an Advanced Optical System" by Kuang et al.**

Kuang et al. presents a theoretical framework for measuring cloud effective supersaturation fluctuations using an advanced optical system, which can improve understanding aerosol activation and cloud microphysics. The framework focuses on observing the critical activation diameter and hygroscopicity of activated aerosols through the scattering and water-induced scattering enhancement of interstitial and activated aerosols. It allows for minute- to second-level effective supersaturation measurements, capturing vital fluctuations for cloud microphysics studies. I think the manuscript, once revised to address the concerns outlined below, could be considered for publication.

**Major comments:**

(1) The theoretical framework introduced in this paper are mainly based on κ-Köhler theory, that is, supersaturation could be obtained with known dry diameter and hygroscopicity kappa. However, the application of κ-Köhler theory is under assumption of water surface tension and fully dissolution. Previous studies have uncovered the surface tension reduction (Gerard et al., 2016; Noziere et al., 2010; Ovadnevaite et al., 2017) and slightly soluble components (Ho et al., 2010) in atmospheric aerosol samples. So, if the apply the framework in field observation, the authors should add some discussion about the uncertainty originated from the above-mentioned assumptions.

(2) As the author mentioned in section 3.2 that the hygroscopicity parameter kappa for supersaturation prediction was $\kappa_{act,f(RH)}$ from aerosol light scattering enhancement factor $f(RH)$ by using humidified nephelometer under unsaturated condition. So, would it bring uncertainty to supersaturation prediction since there may be hygroscopicity deviations between unsaturated and supersaturated condition? Though the authors discussed a bias of 0.1 in $\kappa$ only results in a 0.01% bias in supersaturation retrievals, but the retrievals supersaturation ratio was very low as it was shown in Fig 2c (the lowest value can be 0.02), so 0.01% uncertainty is comparatively large.

(3) As the author mentioned in the paper that we assumed aerosol populations remained unchanged during the 30-minute period (based on comparisons between PM1/PM2.5 and TSP inlets), which can sometimes introduce significant uncertainties in the sizeresolved AR calculations. Based on the authors observation experience, I wonder what is the frequency of the significant uncertainties' events. And 30-minute period was long and the assumption of constant aerosol populations may be not very appropriate, is there any possible improvement to decrease the time period?

(4) As the author mentioned that the supersaturation is effective ratio that make specific number or fraction of aerosol particles activated to CCN, rather than real environment supersaturation ratio. So, I am interested in how to use the "effective ratio" and detect new insight in observation or climate models. Can the author give a simple example or description about it?

**Minor comments:**
(5) Line 199-200: please added some description and references about how to accurately retrieve $D$ by machine learning techniques

(6) References section: The format of the references is not consistent (e.g., some journal names are full but others are abbreviations). Please revised carefully.

**Reference**

Gerard, V., Noziere, B., Baduel, C., Fine, L., Frossard, A. A., & Cohen, R. C. (2016). Anionic, Cationic, and Nonionic Surfactants in Atmospheric Aerosols from the Baltic Coast at Asko, Sweden: Implications for Cloud Droplet Activation. *Environmental Science & Technology, 50*(6), 2974-2982. https://doi.org/10.1021/acs.est.5b05809

Ho, K. F., Lee, S. C., Ho, S. S. H., Kawamura, K., Tachibana, E., Cheng, Y., & Zhu, T. (2010). Dicarboxylic acids, ketocarboxylic acids, α-dicarbonyls, fatty acids, and benzoic acid in urban aerosols collected during the 2006 Campaign of Air Quality Research in Beijing (CAREBeijing-2006). *Journal of Geophysical Research-Atmospheres, 115*(D19), D19312. https://doi.org/10.1029/2009jd013304

Noziere, B., Ekstrom, S., Alsberg, T., & Holmstrom, S. (2010). Radical-initiated formation of organosulfates and surfactants in atmospheric aerosols. *Geophysical Research Letters, 37*. https://doi.org/10.1029/2009gl041683

Ovadnevaite, J., Zuend, A., Laaksonen, A., Sanchez, K. J., Roberts, G., Ceburnis, D., Decesari, S., Rinaldi, M., Hodas, N., Facchini, M. C., Seinfeld, J. H., & O' Dowd, C. (2017). Surface Tension Prevails over Solute Effect in Organic-Influenced Cloud Droplet Activation. *Nature, 546*(7660), 637-641. https://doi.org/10.1038/nature22806

---

## Referee Comment (RC2)

Comments on "Theoretical Framework for Measuring Cloud Effective Supersaturation Fluctuations with an Advanced Optical System"

The authors proposed a theoretical framework to estimate effective supersaturation by measuring the scattering properties of interstitial and activated aerosols. Critical activation diameter (Dd) can be obtained from the measurement/knowledge of scattering coefficient of interstitial aerosols and activated aerosols at three wavelengths, with the knowledge/assumption of aerosol size distribution, size-resolved activation ratio curves, and aerosol composition. The hygroscopicity parameter (κ) can be estimated from the light scattering enhancement factor of activated aerosols based on Kuang et al. (2017). Effective supersaturation can then be calculated from Dd and κ based on κ-Kohler theory. Although I am concerned about the accuracy of the s retrieval, I do think it might be useful to have an instrument to continuously measure the scattering properties of interstitial and activated aerosols and I encourage the authors to develop new instruments to estimate s. I list my major and minor comments below. Hope they are useful to improve the quality of the paper.

Majoy comments:

1. Section 2.1. I think the definition of effective supersaturation is not correct. Line 109~120: "Fluctuations in supersaturation mean that the effective supersaturation, which directly affects aerosol activation, differs from the mean supersaturation." It seems that the authors define effective supersaturation as supersaturation fluctuation. My understanding is that the effective supersaturation mentioned in this study refers to the critical supersaturation corresponding to a certain Da, not supersaturation fluctuation (e.g., due to turbulence). Please clarify the definition of effective supersaturation.

2. Section 3.1 is not well written. For example, Line 177: "As a result, the scattering properties...". Please describe how the scattering properties shown in Fig.1a are calculated. I figure out the answers by reading the caption of Fig.1, later paragraphs, and supplementary materials. Since ACP does not have a page limit, I would recommend the authors add them in the main text to make it easy to read.

3. The AR curve in Fig. 1a is for the average PNSD observed in the North China Plain from six campaigns. Can you measure AR curve at a high temporal resolution? Line 202~203. "For each aerosol size distribution..." What is the temporal resolution of the measured aerosol size distribution? 10 min? Is the temporal resolution of the estimated effective supersaturation limited by the temporal resolution of the measured aerosol size distribution?

Minor comments:

1.  Line 86: "but do not provide precise direct supersaturation measurements." As far as I know, to calculate s based on direct measurements of water vapor pressure and temperature (if possible) is the best way to obtain s. Maybe you want to say "... fluctuations, but they do not provide precise supersaturation measurements at a high liquid water content."

2.  Line 114: change to "S is the saturation ratio over an aqueous solution droplet with a diameter of D,... , ..."

3.  Line 83: change "e" to "water vapor pressure"

---

## Author Response (AR1)

**Dear Editor:**

Thanks for your time. We are grateful for the reviewer's careful inspection of our manuscript, all reviewers have raised some important questions and provided valuable comments and suggestions. All these comments raised by the referees have been explicitly replied point by point and incorporated into the revision. We believe that the revised manuscript is now more convincing than before.

Thank you very much for your attention and consideration.

Sincerely Yours

Ye Kuang

**Responses to anonymous referee #1**

**General comment**:

Kuang et al. presents a theoretical framework for measuring cloud effective supersaturation fluctuations using an advanced optical system, which can improve understanding aerosol activation and cloud microphysics. The framework focuses on observing the critical activation diameter and hygroscopicity of activated aerosols through the scattering and water-induced scattering enhancement of interstitial and activated aerosols. It allows for minute- to second-level effective supersaturation measurements, capturing vital fluctuations for cloud microphysics studies. I think the manuscript, once revised to address the concerns outlined below, could be considered for publication.

**Response**: Thanks for your comments, which really helped improve the manuscript.

**Major Comments**:

**Comment:** The theoretical framework introduced in this paper are mainly based on κ-Köhler theory, that is, supersaturation could be obtained with known dry diameter and hygroscopicity kappa. However, the application of κ-Köhler theory is under assumption of water surface tension and fully dissolution. Previous studies have uncovered the surface tension reduction (Gerard et al., 2016; Noziere et al., 2010; Ovadnevaite et al., 2017) and slightly soluble components (Ho et al., 2010) in atmospheric aerosol samples. So, if the apply the framework in field observation, the authors should add some discussion about the uncertainty originated from the above-mentioned assumptions.

**Response**: Thanks for your comments, we do agree with the surface attention and solubility changes of slight soluble components could impact on the effective supersaturation derivations. The following discussions was added in the discussion part.

"Previous studies have shown that the reduction in surface tension (Nozière et al., 2010;Gérard et al., 2016;Ovadnevaite et al., 2017) associated with surfactants in atmospheric aerosols can affect aerosol activation and, consequently, the derivation of effective supersaturation. However, if the derivation of κ (as done in this study) assumes a constant water surface tension, the impact of surface tension changes is minimized, as these effects are already incorporated in the κ calculation. Nonetheless, differences in surface tension between supersaturated and subsaturated conditions (Davies et al., 2019;Petters and Kreidenweis, 2013), and their impact on effective supersaturation, still exist. Additionally, prior research has suggested that slightly soluble components in aerosols can influence κ values under both supersaturated and subsaturated conditions (Ho et al., 2010;Petters and Kreidenweis, 2008;Lee et al., 2022;Han et al., 2022;Riipinen et al., 2015;Wang et al., 2019). Therefore, κ observed under subsaturated conditions would affect the derivation of effective supersaturation."

**Comment**: As the author mentioned in section 3.2 that the hygroscopicity parameter kappa for supersaturation prediction was $\kappa act, f(RH)$ from aerosol light scattering enhancement factor $f(RH)$ by using humidified nephelometer under unsaturated condition. So, would it bring uncertainty to supersaturation prediction since there may be hygroscopicity deviations between unsaturated and supersaturated condition? Though the authors discussed a bias of 0.1 in $\kappa$ only results in a 0.01% bias in supersaturation retrievals, but the retrievals supersaturation ratio was very low as it was shown in Fig 2c (the lowest value can be 0.02), so 0.01% uncertainty is comparatively

large.

**Response**: Thanks for your comment, we agree with that 0.01% is comparatively large when average SS is also low. We replotted Figure S1 as the following:

[Figure]

It could be found that 0.1 bias in $\kappa$ result different SS bias at different SS levels, and the SS bias would increase as SS increase. we revised the sentence in the discussion part as:

"Additionally, $\kappa$ measured under subsaturated conditions differs from that under supersaturated conditions (Tao et al., 2023) might also bring some uncertainties. However, as shown in Fig. S1b, even a bias of 0.1 in $\kappa$ only result in a ~0.01% bias when SS is ~0.1% and a ~0.005% bias when SS is ~0.05% in supersaturation retrievals, making the first-order estimates of $\kappa_{D_a}$ from optical measurements generally suitable for supersaturation observations."

Also, a comprehensive uncertainty analysis after this part of discussion was added in Sect.4 in the revised manuscript.

**Comment**: As the author mentioned in the paper that we assumed aerosol populations remained unchanged during the 30-minute period (based on comparisons between PM1/PM2.5 and TSP inlets), which can sometimes introduce significant uncertainties in the size resolved AR calculations. Based on the authors observation experience, I wonder what is the frequency of the significant uncertainties' events. And 30-minute

period was long and the assumption of constant aerosol populations may be not very appropriate, is there any possible improvement to decrease the time period?

**Response**: Thanks for your comment, this is a good question. For the observed radiation fog case in this study, the assumption that aerosol populations remained unchanged is generally applicable because the stagnant characteristics as demonstrated by Kuang et al. (2024) . However, as we recently observed in mountain clouds, aerosol population might have already changed in minute level because the quick movement of air parcels, which means wind speed is a key factor, and the spatial inhomogeneous of aerosol distribution is another important factor. In this year, we have decreased the time of observing size-resolved AR to about to 10 minutes interval based on that scanning period of SMPS is 5 minutes. The time resolution of size-resolved AR measurements depends on the time resolution of aerosol size distribution measurements.

**Comment**: As the author mentioned that the supersaturation is effective ratio that make specific number or fraction of aerosol particles activated to CCN, rather than real environment supersaturation ratio. So, I am interested in how to use the "effective ratio" and detect new insight in observation or climate models. Can the author give a simple example or description about it?

**Response**: Thanks for your comment, I am also exploring this issue. My conclusion is that supersaturation fluctuations caused by turbulence and other factors would made the physically mean supersaturation not application in quantification of aerosol activation, as the existence of so-called sub-saturation cloud caused by turbulence. However, the

key part of Twomey effect is that cloud number difference, which means determining cloud number is one crucial part (might be the most important part, because the concept of water vapor competition is based on number difference), and the growth of cloud droplets is another crucial part. In view of this, accurately determining effective supersaturation not mean supersaturation is very important in climate models with respect to aerosol activation, however, current models are describing mean supersaturation, we need have the capabilities to effectively observe effective supersaturation first.

We also added more discussions in Sect.4 to manifest the potential applications of effective supersaturation fluctuation measurements.

"As mentioned in Sect. 2.1, the theoretical framework proposed in this study is designed to observe effective supersaturation fluctuations, rather than supersaturation fluctuations themselves. While there are non-negligible uncertainties associated with observing effective supersaturation using the proposed theory, the size and hygroscopicity distributions of total interstitial and activated aerosol populations remain nearly constant when measured with second-scale or shorter time resolution. The parameter that changes over time is the dynamic exchange between interstitial and activated aerosols. Consequently, fluctuations in the scattering signals of interstitial and activated aerosols can reflect this exchange at high temporal resolution. Since effective supersaturation fluctuations result from underlying supersaturation variations, they could, in principle, provide insights into the causes of these fluctuations, such as turbulence, though this would require further investigation and endeavor. In addition, for size-resolved AR, both $\sigma$ and MAF are crucial parameters. However, using scattering coefficients at just three wavelengths of Aurora 3000 nephelometer is insufficient for accurately retrieving $\sigma$ and MAF. If $\sigma$ and MAF could be measured more precisely through the extended optical framework, it would provide deeper insights into supersaturation fluctuations."

**Minor Comments**:

**Comment**: Line 199-200: please added some description and references about how to accurately retrieve $D$ by machine learning techniques.

**Response**: Thanks for your comment, we have added details of the machine learning procedure in Sect 3.1.

In the manuscript, the following paragraph was added:

"This assumption was tested using Mie theory, based on aerosol size distributions sampled during six campaigns conducted in the North China Plain region (Kuang et al., 2018). For each aerosol size distribution, we randomly assumed different activation curves using Eq.2. That is, for each PNSD from those campaigns, the scattering coefficients of submicron interstitial and activated+interstital aerosols at wavelengths of 450 nm, 525 nm and 635 nm corresponding to nephelometer case under 100 size-resolved AR scenarios were simulated using the procedure. And each size-resolved AR curve was produced by using randomly produced $D_a$, $\sigma$ and MAF as inputs of Eq.2. In the random step, the range of $D_a$ is 100-700 nm, the range of $\sigma$ is 1-30, the range of MAF is 0.5-1. In each pair, simulated $\sigma_{sp,PM_1,inter}(dry,\lambda)$—at 450 nm, 525 nm, 635 nm, and $\sigma_{sp,PM_1,all}(dry,\lambda)$ at 450 nm, 525 nm, 635 nm was the x values of the random forest model, corresponding $D_a$ is the y value of the random forest model, and the random forest package from Python Scikit–Learn machine learning library (http://scikit-learn.org/stable/index.html) is used for this purpose. With these configurations, more than million pairs are simulated. To preliminarily validate this approach, we randomly selected 75% of the simulated data pairs for training the model, while the remaining 25% were used for validation."

**Comment**: References section: The format of the references is not consistent (e.g., some journal names are full but others are abbreviations). Please revised carefully.

**Response**: Thanks for your comment, abbreviations were revised as full names.

**Responses to anonymous referee #2**

**General comments**:

The authors proposed a theoretical framework to estimate effective supersaturation by measuring the scattering properties of interstitial and activated aerosols. Critical activation diameter (Dd) can be obtained from the measurement/knowledge of scattering coefficient of interstitial aerosols and activated aerosols at three wavelengths, with the knowledge/assumption of aerosol size distribution, size-resolved activation ratio curves, and aerosol composition. The hygroscopicity parameter (κ) can be estimated from the light scattering enhancement factor of activated aerosols based on Kuang et al. (2017). Effective supersaturation can then be calculated from Dd and κ based on κ-Kohler theory. Although I am concerned about the accuracy of the s retrieval, I do think it might be useful to have an instrument to continuously measure the scattering properties of interstitial and activated aerosols and I encourage the authors to develop new instruments to estimate s. I list my major and minor comments below. Hope they are useful to improve the quality of the paper.

**Response**: Thanks for your comment, we have improved the manuscript based on your suggestions, also more discussions about accuracy of effective supersaturation measurements using the proposed framework was added in the revised manuscript. We are now working on developing an instrument that could achieve continuous measuring of the scattering and hygroscopic properties of interstitial and activated aerosols as well as the effective supersaturation based on the proposed framework. We also added more discussions in Sect.4 to manifest the potential applications of effective supersaturation

fluctuation measurements.

**Major Comments:**

**Comment**: Section 2.1. I think the definition of effective supersaturation is not correct. Line 109~120: "Fluctuations in supersaturation mean that the effective supersaturation, which directly affects aerosol activation, differs from the mean supersaturation." It seems that the authors define effective supersaturation as supersaturation fluctuation. My understanding is that the effective supersaturation mentioned in this study refers to the critical supersaturation corresponding to a certain Da, not supersaturation fluctuation (e.g., due to turbulence). Please clarify the definition of effective supersaturation.

**Response**: Thanks for your comment. Yes, the effective supersaturation mentioned in this study refers to the critical supersaturation corresponding to a certain Da, not supersaturation fluctuation. The sentence "Fluctuations in supersaturation mean that the effective supersaturation, which directly affects aerosol activation, differs from the mean supersaturation" try to explain that the supersaturation fluctuations would result in the difference between effective supersaturation with mean supersaturation. To make this clear, this paragraph is revised as the following:

"The concept of effective supersaturation was introduced based on aerosol activation measurements (Hudson and Yum, 1997;Hudson et al., 2010), which could be defined as the supersaturation in CCN chamber (CCN activation under constant supersaturation conditions) that resulted in the same aerosol activation fraction with the observed aerosol activation fraction in clouds. Quick fluctuations in supersaturation would result in the effective supersaturation, which directly determined by aerosol activation, differs from the mean supersaturation which is determined by average water vapor content and temperature. However, the concept of κ-Köhler theory is established according to a constant

supersaturation scenario, therefore provides a framework for deriving effective supersaturation from aerosol activation measurements in clouds (Petters and Kreidenweis, 2007)."

**Comment**: Section 3.1 is not well written. For example, Line 177: "As a result, the scattering properties…". Please describe how the scattering properties shown in Fig.1a are calculated. I figure out the answers by reading the caption of Fig.1, later paragraphs, and supplementary materials. Since ACP does not have a page limit, I would recommend the authors add them in the main text to make it easy to read.

**Response**: Thanks for your comment, we have added this part as Sect.2.3 in the revised manuscript.

**Comment**: The AR curve in Fig. 1a is for the average PNSD observed in the North China Plain from six campaigns. Can you measure AR curve at a high temporal resolution? Line 202~203. "For each aerosol size distribution…" What is the temporal resolution of the measured aerosol size distribution? 10 min? Is the temporal resolution of the estimated effective supersaturation limited by the temporal resolution of the measured aerosol size distribution?

**Response**: Thanks for your comment, the AR curve in Fig.1a is just an example to demonstrate the remarkable difference in scattering properties of interstitial and activated aerosols in clouds. As the response to reviewer#1 that we could observe the AR curve in a high temporal resolution if we have a instrument that could measure aerosol size distribution in high temporal resolution, for example, minutes level. The temporal resolution of measured aerosol size distribution in the six campaigns is 5

minutes. However, the temporal resolution of PNSD measurements in these campaigns is not important in the machine learning prediction training for Da prediction (the shape of PNSD matters). We use PNSD measurements from six campaigns is to include as many PNSD shapes as possible, therefore, PNSD measurements in these campaigns were averaged to 30 minutes resolution to reduce simulation time of the training datasets. The temporal resolution of estimated effective supersaturation depends on time resolution of measured scattering signals of interstitial and activated aerosols in clouds, not the time resolution of training data.

**Minor Comments:**

**Comment**: Line 86: "but do not provide precise direct supersaturation measurements." As far as I know, to calculate s based on direct measurements of water vapor pressure and temperature (if possible) is the best way to obtain s. Maybe you want to say "… fluctuations, but they do not provide precise supersaturation measurements at a high liquid water content."

**Response**: Thanks for your comment, we agree that calculating SS based on direct measurements of water vapor pressure and temperature is best way to obtain SS if possible, however, it is almost impossible with current techniques. What I want to say is that previous studies used this way to estimate supersaturation fluctuations, however only fluctuations, the mean supersaturation could not be accurately measured. To make this clear, we revised this sentence as:

"Although current techniques of water vapor and temperature measurements could not achieve accurately measurements of supersaturation, however, direct measurements of water vapor pressure

and temperature were previously used to estimate supersaturation fluctuations"

**Comment**: Line 114: change to "S is the saturation ratio over an aqueous solution droplet with a diameter of D,… , …"

**Response:** revised

**Comment**: Line 83: change "e" to "water vapor pressure"

**Response**: Revised

**Responses to anonymous referee #3**

**General comment**:

This paper proposes a method for determining the supersaturation in a fog or cloud. The topic is highly important due to the crucial role played by supersaturation in determining the concentration and shape of the cloud droplet size distribution. The measurement concept seems novel and robust, and is reasonably well explained in the paper. There are a few places that are not clear, which I note below. I'm not sure why this was submitted to ACP instead of AMT, but that's for the editors to decide regarding appropriateness. I consider the manuscript suitable for publication after the following comments are addressed.

**Response**: Thanks for your comments, the key reason that we decided to submit this manuscript to ACP not AMT is because that this paper is not about instrument development, but about the concept and theoretical framework behind the effective supersaturation measurements using advanced optical systems, and the system could be achieved based on different ways, for example, totally different optical sensors. Therefore, this manuscript is more about physics not techniques.

**Major Comments:**

**Comment**: One key question that arises after reading the paper is how the proposed technique can provide information about supersaturation fluctuations versus mean

supersaturation. The authors nicely describe the importance of quantifying fluctuations, but then it is not clearly explained later how this can be accessed. I believe it is related to the sigma value in Equation 2 (see my comment below) but I did not find a discussion of this topic in the paper, except a brief mention in the Discussion (limitation 1). Please provide more discussion of this topic.

**Response:** Thanks for your comment, this comment really helped us a lot to add further clarification and refinement.

We added the following paragraph in Sect.2.1 to make it clear what types of supersaturation fluctuations could be measured using the proposed framework:

"Two types of supersaturation fluctuations have been previously identified. The first type involves fluctuations in supersaturation directly governed by water vapor pressure and temperature, as described by Siebert and Shaw (2017). These fluctuations are linked to turbulence and water phase changes that influence water vapor pressure and temperature. The second type concerns fluctuations in effective supersaturation, which are associated with the activation and deactivation processes of aerosols, as noted by Ditas et al. (2012). The first type of fluctuations dictates the instantaneous growth and evaporation of droplets, thereby controlling the activation and deactivation of cloud droplets. As such, the second type of fluctuation is inherently driven by the first type. The theoretical framework proposed in this study enables the measurement of fluctuations in effective supersaturation."

We added the following paragraph in the discussion part to demonstrate the potential applications of the measured effective supersaturation fluctuations:

"As mentioned in Sect. 2.1, the theoretical framework proposed in this study is designed to observe effective supersaturation fluctuations, rather than supersaturation fluctuations themselves. While there are non-negligible uncertainties associated with observing effective supersaturation using the proposed theory, the size and hygroscopicity distributions of total interstitial and activated aerosol

populations remain nearly constant when measured with second-scale or shorter time resolution. The parameter that changes over time is the dynamic exchange between interstitial and activated aerosols. Consequently, fluctuations in the scattering signals of interstitial and activated aerosols can reflect this exchange at high temporal resolution. Since effective supersaturation fluctuations result from underlying supersaturation variations, they could, in principle, provide insights into the causes of these fluctuations, such as turbulence, though this would require further investigation and endeavor. In addition, for size-resolved AR, both σ and MAF are crucial parameters. However, using scattering coefficients at just three wavelengths of Aurora 3000 nephelometer is insufficient for accurately retrieving σ and MAF. If σ and MAF could be measured more precisely through the extended optical framework, it would provide deeper insights into supersaturation fluctuations."

**Specific comments:**

**Comment**: Lines 84-85: the statement "estimated from vertical velocity measurements" would be better supported by citing a paper that uses that approach, such as the paper by Cooper 1989 (J. Atmos. Sci.). Also, it would be more correct to state "estimated from vertical velocity and droplet size distribution measurements.

**Response**: The reference is added, and the sentence is revised accordingly.

**Comment**: Lines 117-119: "The κ-Kohler theory tells that if the critical diameter of aerosol activation ($D\_a$) and corresponding aerosol hygroscopicity parameter κ are known, the surrounding supersaturation can be retrieved based on air temperature measurements and by assuming $\sigma\_s$\a the surface tension of water." This statement is correct, but it assumes knowledge that is not stated, such as how kappa and the critical

diameter are related to each other. Please explain more thoroughly.

**Response**: Thanks for your comment, the following discussions are added after this sentence to explain more:

"Note that $D_a$ and $\kappa$ are not independent with each other, average $\kappa$ of aerosols with diameter $D_a$ is needed. Previous studies have shown that the reduction in surface tension (Nozière et al., 2010;Gérard et al., 2016;Ovadnevaite et al., 2017) associated with surfactants in atmospheric aerosols can affect aerosol activation and, consequently, the derivation of effective supersaturation. However, if the derivation of $\kappa$ (as done in this study) assumes a constant water surface tension, the impact of surface tension changes is minimized, as these effects are already incorporated in the $\kappa$ calculation. Nonetheless, differences in surface tension between supersaturated and subsaturated conditions (Davies et al., 2019;Petters and Kreidenweis, 2013), and their impact on effective supersaturation, still exist. Additionally, prior research has suggested that slightly soluble components in aerosols can influence $\kappa$ values under both supersaturated and subsaturated conditions (Ho et al., 2010;Petters and Kreidenweis, 2008;Lee et al., 2022;Han et al., 2022;Riipinen et al., 2015;Wang et al., 2019). Therefore, $\kappa$ observed under subsaturated conditions would affect the derivation of effective supersaturation."

**Comment**: Line 126: In describing Equation 2 it is stated that "$\sigma$ is associated with the slope of the curve near D_a". Please provide a physical interpretation of what factors contribute sigma. Would it be true that for a monodisperse aerosol, and uniform supersaturation, sigma would be zero?

**Response**: Thanks for your comment, this sentence is revised as:

"$\sigma$ is associated with the slope of the size-resolved AR curve near $D_a$ and mostly influenced by the heterogeneous distribution of aerosols near $D_a$ as well as supersaturation fluctuations (note that not effective supersaturation fluctuations)"

**Comment**: Lines 134-136: This part of the sentence is not clear and should be revised: "which brings uncertainty in $Da$ derivations due to that the maximum activation fraction of aerosols larger than $Da$ does not equal to unit although usually very close to (Tao et al., 2018b).

**Response**: Thanks for your comment, this sentence was revised as:

"which brings uncertainty in $D_a$ derivations due to that not all aerosols larger than $D_a$ are activated, because the MAF in Eq.2 does not equal to unit although usually very close to (Tao et al., 2018)"

**Comment**: Figure 3: Include brief discussion in the figure caption to explain the underlying concept of the instrument, at least for panel a. For example, explain the purpose of the drier versus the cooler (for controlling humidity).

**Response**: Thanks for your comment, following sentences are added to explain:

"The heater upstream of the sample is used to reduce the relative humidity (RH) to below 60%, ensuring the evaporation of most of the water content, to make sure the consistency of needed PM$_1$ cut. The cooler upstream of the 'wet' nephelometer increases the sample RH to approximately 90%, allowing hygroscopicity measurements under conditions close to supersaturation."

**Comment:** Discussion: some discussion of the expected precision versus accuracy, as well as estimated uncertainties should be included.

**Response**: Thanks for your comment, the following paragraph was added in Sect.4:

[revised manuscript text omitted]